# Seedling Characteristics of Three Oily Species before and after Root Pruning and Transplant

**DOI:** 10.3390/plants8080258

**Published:** 2019-07-30

**Authors:** Ofelia Andrea Valdés-Rodríguez, Arturo Pérez-Vázquez

**Affiliations:** 1Academia de Desarrollo Regional Sustentable El Colegio de Veracruz, Xalapa 91000, Mexico; 2LPI3 Colegio de Postgraduados, Predio Tepetates, Veracruz 91690, Mexico

**Keywords:** *Moringa oleifera*, *Jatropha curcas*, *Ricinus communis*, root systems

## Abstract

*Moringa oleifera* Lam. (*Moringa*), *Jatropha curcas* L. (*Jatropha*), and *Ricinus communis* L. (*Ricinus*) are oily species known by their capability to grow in tropical and subtropical lands. However, there are no studies comparing their growth and recovery capabilities after root pruning and transplant. The purpose of this research was to compare and analyze propagation, growth, and recovery performance of these species after root pruning and transplant. We sowed 100 seeds per species and monitored their survival and growth during a 63-day period; after this, we uprooted the plants and pruned their roots 4.0 cm from their base and transplanted them. We monitored their recovery over 83 days, and then uprooted plants and measured above- and belowground data, digitized their roots in three dimensions, and calculated biomass fractions. With this information, we established allometric equations to estimate biomass fractions and root distribution models. Results indicated that *Ricinus* had the highest propagation capabilities. *Jatropha* and *Ricinus* had similar recovery after root pruning and transplant. *Moringa* had the lowest propagation and recovery from transplant. Concerning belowground data, root pruning increased root density more than three times in *Moringa*, four times in *Ricinus*, and six times in *Jatropha*. Nevertheless, the three species maintained natural root trays. *Ricinus* had the longest and thinnest roots and the highest number of branches, followed by *Jatropha*, and finally *Moringa*, with the smallest quantity and the shortest and thickest roots. We concluded that the three species recovered well from root pruning and transplant, with improved root structure upon applying these practices.

## 1. Introduction

*Moringa oleifera* Lam. (*Moringa*), *Ricinus communis* L. (*Ricinus*), and *Jatropha curcas* L. (*Jatropha*) are oily species recognized by their multiple industrial applications [1,2,3]. In their original region, *Moringa* trees are typically found growing in areas with poor sandy soils and at altitudes below 1400 m above sea level (masl), where it can survive temperatures up to 40 °C and minimal annual rainfalls of 250 mm [1,4]. *Ricinus* is found in tropical and subtropical lands, where precipitations can be as low as 400 mm per year and with altitudes going from zero up to 2300 masl [5]. *Jatropha* is considered to be native to America, where it is found in tropical and subtropical climates with precipitation ranges of 800–3000 mm/year, diverse soil types, temperature ranges of 10.5–32 °C, and altitudes up to 1000 masl [6,7]. However, there is no information comparing their capability to survive and develop under the same environmental conditions and under stressful situations, such as transplant shocks. This data is important if these species are to be sown intercropped or when one of them, among the others, is recommended as an alternative new crop. Another aspect to consider is that for their establishment, direct sowing requires higher investment in land preparation and to take care of the seedlings at the field. Therefore, the most common practice consists of propagating seedlings under greenhouse conditions and then transplanting them to the field. This method regularly requires roots to be extracted out of their container, where a fraction of the roots are cut to avoid bending [8]. Although there are not specific recommendations for how to trim the roots of these species, it is known that this practice induces a “transplant shock”, which is a condition of distress caused by root injuries and the need for the plant to adapt to a new environment [9]. Besides this, it has been reported that root pruning can decrease vegetative growth rates in trees and crops of different species but, at the same time, it can stimulate the development of new roots, contributing to the improvement of root biomass [10,11].

Root biomass and root morphology determine the capacity of the plant to capture nutrients and water, affecting its survival and development. Roots also provide support to plants; consequently, a strong root system with a good morphology will facilitate plant anchorage against uprooting forces, like strong winds or storms [12]. Therefore, learning about stem and root development after trimming and transplant is an important knowledge for determining the key factors for successful establishment on the field [13], and to make decisions over which crop is more appropriate for specific situations. Hence, studying propagation, growth patterns, and responses to root pruning and transplant of *Jatropha*, *Moringa*, and *Ricinus*, will allow us to understand their natural capabilities and their particular strategies to recover from transplant shock.

We also have to consider that root trimming can change the original root structure of the plant; therefore, relationships between above- and belowground data will be affected. Even though there are studies involving above- and belowground parameters of these three species [14,15,16,17], none of them evaluates plant characteristics and their performance before and after trimming of roots and transplant.

Previous research from these species indicates that during its juvenile stage, *Moringa* develops a napiform tap root from which very thin second-order roots emerge [18], *Jatropha* develops a thick tap root with mainly four thick laterals and few second-order roots [16], and *Ricinus* develops a tap root with many thin laterals and second-order roots [19]. The main function of the tap root is to provide anchorage, while laterals and second-order roots increase stabilization and the search for resources [20]. Thus, if we prune the main roots, plants would have to adapt their structure to preserve their original functions. We theorized that the tap root may develop new vertically oriented roots to provide anchorage, while new second-order roots will emerge from the trimmed laterals to maintain stability and search for more resources. Yet, plant responses will depend on the specific strategy of each species to survive and recover from transplant shock. However, there are no data of pruned root development for any of these plants, nor there are models linking above- and belowground biomass fractions after root pruning and transplant. Thus, we want to determine how root trimming affects plant morphology and the key differences of each species representing above- and belowground relations prior to and post-transplant, in order to understand how each one deals with the shock. Our specific questions were: Which is the species with the highest propagation capabilities? How does each species respond to root trimming and transplant? Which one has the fastest recovery process? Does root pruning affect the natural below- and aboveground structures?

Therefore, the aims of this research were 1) to evaluate propagation and growth characteristics of *Moringa*, *Ricinus*, and *Jatropha* in a subtropical climate, 2) compare their above- and belowground relationships before and after root pruning, and 3) analyze their root structure by modeling root biomass distribution and above- and belowground relationships after pruning to compare the plant capabilities of each species.

## 2. Materials and Methods

### 2.1. Plant Material

All plant material was obtained from Mexican collections. *Ricinus* seeds were collected at the Mexican plateau in the state of San Luis Potosi (22.191744° N, −101.010241° W; semi-dry semi-warm climate—Bsh) during June; average seed weight (± standard deviation) was 487 ± 52 mg. *Moringa* seeds came from the state of Morelos (18.523889° N, −98.790278° W; sub-humid warm climate—Aw_1_), where they were collected during August; average seed weight (± standard deviation) was 310 ± 74 mg. *Jatropha* seeds were collected in Alvarado, Veracruz (18.79386° N, −95.79886° W; humid-warm climate—Aw_2_) during September; average seed weight (± standard deviation) was 686 ± 89 mg. All seeds were stored in a refrigerator at −5 °C prior to their use on the first day of October. For each species, 100 seeds were selected. We selected seeds after considering if they were free from damage or scratches.

### 2.2. Environmental Conditions and Substratum

The experiment was established at the facilities of Colegio de Postgraduados, Campus Veracruz, which is located in the municipality of Manlio Fabio Altamirano, Veracruz (19°16′00” N, 96°16′32” W, 18 masl). The climate at the site is considered sub-humid warm (Aw_1_). The experimental site was located under a square sun mesh providing 50% shading. The sun mesh was supported by poles and open by the four sides. During the experiment, environmental temperature and humidity of the site were registered by the local climatic station. The substrate used for this experiment consisted in a homogeneous mix of 5:1 sand (collected from a natural dune) and compost made of sugarcane bagasse and soil of the region (sand/compost 5:1). Physical and chemical analysis of separate samples of sand and compost were previously performed by the soil laboratory of Colegio de Postgraduados. The granulometric characterization (Bouyoucos method) for sand indicated 96% sand, 2.5% silk, and 1.5% clay; compost consisted of 52% sand, 28% silk, and 20% clay; pH (electronic potentiometer) of 7.03 for sand and 7.01 for compost; nitrogen (N) (Kjeldahl method) not detected for sand and 1.05% for compost.

### 2.3. Germination and Survival

This experiment consisted of a random design with 100 seeds per species. Each seed was placed inside a plastic bag of 20 cm length and 15 cm wide. The bag was filled with sand/compost 5:1 substratum. Bags were labeled for identification and randomly distributed over three tables (1.2 m height). Seeds and seedlings were watered every morning until excess water drained away during a 63-day period.

The number of germinated and survival plants were registered daily. The percentage of germination was estimated on day 63 as the number of germinated seedlings over the number of sowed seeds. The mean germination time in days was calculated as the time when 50% of seeds have germinated.

### 2.4. Aboveground Growth before Transplant

The experiment consisted of a random design with 100 seedlings per species. Seeds that did not germinate from the previous experiment were replaced with germinated seeds from storage. For each seedling, we established an independent register to maintain its age corresponding to a fixed period of measurement, which was set every seven days. We measured plant height (as the length from the base of the stem to the highest leaf tip), root collar diameter (RCD) (using a digital caliper with 0.01 mm precision), leaf length and width of the youngest fully mature leaf (using a tape measure with 1.0 mm precision), and we counted the number of leaves. The leaf area of the youngest fully mature leaf was estimated for *Jatropha* as in Liv et al. [21], for *Ricinus* as in Jain and Misra [22], and for *Moringa*, the compound leaf area was estimated as in Valdés-Rodriguez [14]. Each plant was measured until the 63rd day after sprouting.

### 2.5. Leaf Thickness and Weight Measurements

Six healthy leaves of each species were randomly selected and measured by their blade length, width (using a tape measure, 1 mm precision), and thickness, without veins and including veins (using a digital caliper, 0.01 mm precision); petiole length, thickness at the base of the bud, and thickness at the base of the blade. Following this, leaves were divided in blade and petiole and weighed (using an analytical balance with 0.0001 g precision). For *Moringa*, only one leaflet of the compound leaf was selected.

### 2.6. First Uprooting and Belowground Measurements

Plants were uprooted when they were 63 days old. Root systems were measured as follows: length of tap and lateral roots using a tape measurement with 1 mm precision; diameter of tap and lateral roots at the base and at the tip of the roots using a digital caliper with 0.01 mm precision, and number of lateral roots. We labeled tap root as the one emerging from the center of the stump of the plant, and lateral roots as the ones emerging from the main trunk. For each root system, we estimated the volume of tap and lateral roots as in Valdés et al. [16].

### 2.7. Root Pruning and Transplant

The roots of the uprooted plants were gently cleaned to remove excess of soil in order to observe the bare roots, and all roots were cut at a distance of 4.0 cm from the base. We then transplanted each plant in a plastic bag of 50 cm length and 30 cm width filled with the substratum (sand/compost 5:1) and watered each bag until field capacity. After this procedure, we watered the plants every two days during a period of 83 days.

### 2.8. Aboveground Measurements after Transplanting

We waited for a period of 30 days post-transplant before measuring plant height, root collar diameter, leaf area, and the number of leaves, branches, flowers, and fruits in individuals that presented these parts. Measurements were then taken about every two weeks for a period of 53 days.

### 2.9. Second Uprooting and Belowground Measurements

At 83 days post-transplanting, 20 plants per species were randomly selected and uprooted again, and their root systems were measured as in the first uprooting. Additionally, the root structure of one plant from each species was digitized as in Valdes et al. [10]. Digitized roots were projected in a three-dimensional scheme by using the program Visualea [23] to obtain a graphical representation of the effects that trimming of roots had had over root systems.

#### 2.9.1. Root Distribution Models

For each species, roots thicker than 3.0 mm were counted and measured by length, thickness, and angle of inclination along horizontal and vertical axes every 50 mm, with the help of macros implemented in Excel 2010^®^ [14,16]. After this, based on the projected roots, we tested logistic and sigmoid models available at the fit regression tool of SigmaPlot 10.0 to obtain representative models of root volume distribution. The model with the highest coefficient of determination (r^2^) for each species was selected to represent and compare root volume distribution among species.

#### 2.9.2. Biomass Estimation by Allometric Relationships

Biomass estimation was determined by considering the highest correlation between the allometric data (plant height, RCD, number of leaves, and leaf area) and leaf, stem, and root biomass. After this, we tested linear, logarithmic, and power regression models for biomass estimation [24,25,26]. The best model was selected considering the highest r^2^ of each estimated variable.

### 2.10. Statistical Analysis

Comparisons of germination and survival among species were performed by a log-rank test and the Kaplan–Meier method with a significance level of 0.01 and using SigmaPlot 10.0 software. Since we only tested 100 seeds per species, these germination results were compared with previous experiments reported for the same genotypes of these species at the same site. This way, we presented a discussion over propagation capacities of each species at this site.

For the first growing phase (previous to the first uprooting), we compared aboveground parameters per each sampling period, while for the second growing phase (post-transplant), we compared only the last measurement (83 days post transplanting). Since all the plants were pruned and transplanted, post-transplant data were compared with plants of the same genotype that had been grown without transplanting during previous experiments at the same site.

Above- and belowground relationships were evaluated by the Pearson product moment correlation, and relationships between each two pair of variables were evaluated at 0.05 and 0.01 significance levels.

Comparisons between species were conducted by one-way analysis of variance (ANOVA). For each group of data, normality and equal variance tests were performed before ANOVA; therefore, if any of these tests failed, the non-parametric Kruskal–Wallis method was used, otherwise the parametric ANOVA was used. *Post hoc* tests of paired samples were accomplished by the Tukey method when samples passed normality and equal variance tests; otherwise the non-parametric Dunn’s method was used. All comparisons were performed at a 0.05 significance level. SigmaPlot 10.0 software was used for all analyses.

## 3. Results

### 3.1. Germination and Survival

Germination curves were different among species (P < 0.001), *Ricinus* had the fastest germination time and the highest germination percentage, followed by *Jatropha* and *Moringa*. These results, and data from previous experiments performed in this site with the same genotypes, are presented in Table 1.

### 3.2. First Phase of Seedling Growth

#### 3.2.1. Aboveground Measurements

Statistical analysis indicated that *Ricinus* was taller than *Moringa* and *Jatropha* since day 14 (P < 0.001), while *Jatropha* was taller than *Moringa* until day 35 after sprouting (P < 0.001). By day 42 *Moringa* and *Jatropha* registered similar heights (P = 0.11); however, by day 63, *Moringa* was taller than *Jatropha* (Figure 2). *Jatropha* had the highest root collar diameter since day seven after sprouting (P < 0.001). *Jatropha* always had the lowest relationship between height and RCD (slender index); while *Ricinus* had the highest slender index since day 49 after sprouting until the end of the experimental period. Plant growth rates and a comparison with previous data from the same experimental site are shown in Table 2.

#### 3.2.2. Leaf Measurements

*Moringa* had the highest number of leaves since day 35 (P < 0.001), while *Ricinus* had the lowest number of leaves since day 49 after sprouting. Leaf area was different among species (P < 0.001) and *Ricinus* had the highest leaf area. *Jatropha* registered the thicker blades of the three species (P < 0.05), with 0.26 mm, followed by *Ricinus* with 0.24 mm, while *Moringa* had the thinnest blades at 0.23 mm. *Ricinus* had the heaviest blades (4.1 g), followed by *Jatropha* (3.2 g), and *Moringa* had the lightest (0.03 g).

### 3.3. Relationships between Above- and Belowground in 63-Day-Old Seedlings

Root size and morphology were statistically different among species (Table 3, Figure 1). *Ricinus* had the longest and thinnest roots, while *Moringa* had the thickest tap root and the lowest number of lateral roots.

Allometric relationships between above- and belowground data indicated that *Jatropha* had the longest roots in relation with the length of its stem and the highest correlation between RCD and root volume. *Moringa* had the shortest roots in relation to its height and the highest correlation between height and root volume, while *Ricinus* had the highest correlation between root volume and leaf area (Table 4).

### 3.4. Second Phase: Plant Performance after Transplanting

#### 3.4.1. Aboveground Measurements

The three species registered a 100% survival rate after transplanting. Nevertheless, transplant caused the loss of 18% of the leaves of *Moringa* and 5.0% of *Ricinus*, but defoliation was not observed for *Jatropha*. At the end of the experimental period, we recorded the highest leaf area of the first mature leaf in *Ricinus*, being 1.5 times bigger than *Jatropha* and 2.0 times bigger than *Moringa*.

The highest stem lengths were registered in *Ricinus*, and the thickest stems were recorded in *Jatropha* (Figure 2). *Ricinus* was the only species to reach maturity, with 12% of its plants flowering 46 days after transplant and 45% of individuals flowering and fruiting 142 days after sprouting. In order to compare results, data from non-transplanted plants registered from previous experiments at the same site are shown in Table 5.

#### 3.4.2. Above- and Belowground Relationships

Belowground measurements are shown in Table 6, while the correlation coefficients between above- and belowground data are shown in Table 7. Figure 3 shows the labeling identification of each measurement. The “trimmed tip” was identified as the limit where roots were cut before transplanting. *Ricinus* was the species with the highest root elongation before and after trimming, and with the highest number of lateral roots, but it was also the species with the thinnest roots. *Moringa* had the lowest number of lateral roots and the shortest roots, but it has the thickest roots of the three species.

Figure 4 shows the digitized roots of *Jatropha*, *Moringa*, and *Ricinus*. For the three species, trimming of tap roots induced the formation of new root branches at the trimmed limbs. The new root branches emerging from the tap root had average inclinations of −82° for *Ricinus* and *Jatropha*, and −60° for *Moringa*.

#### 3.4.3. Root Distributions and Their Models

Root volume distribution indicated that *Moringa* had the highest volume distributed around the first 10 cm of its stem, although its root volume decreased rapidly along horizontal and vertical surfaces (Figure 5). *Jatropha* and *Ricinus* had their highest volume distributed along the first 5 cm around the stump. *Ricinus* had the lowest root volume decrement along its vertical and horizontal distribution. *Moringa* the thickest tap root and the highest number of roots with the thickest diameters, but its roots were the shortest of the three species. *Ricinus* had the highest number of roots with the smallest diameters and the longest roots along vertical and horizontal surfaces.

Root distribution models were best fitted by logistic models for the vertical distribution and by exponential models for the horizontal distribution. The parameters obtained for each model and their corresponding r^2^ and square sums (SS) are shown in Table 8.

### 3.5. Plant Fresh and Dry Weight

*Moringa* had the heaviest fresh and dry roots of the three species, while *Jatropha* had the heaviest fresh and dry stem and total weight of the three species (P < 0.05). *Ricinus* had the lightest fresh and dry roots of the three species, as it had the lightest total weight of the three species (Table 9).

### 3.6. Allometric Relationships for Above- and Belowground Biomass Estimation

Table 10 shows the models with the highest coefficients of regression between allometric data and above- and belowground biomass estimation.

## 4. Discussion

### 4.1. Germination and Survival

Results from this experiment indicated that *Ricinus* can propagate faster and had a greater survival than *Jatropha* and *Moringa* in these subtropical conditions (warm sub-humid climate and 5:1 sand/compost substratum). In this regard, *Ricinus* was the most adaptable species, with higher plasticity to respond to different environmental conditions [31], which is confirmed by the fact that these seeds came from a drier site (Bsh climate), and they had 100% germination rate in a higher environmental humidity (sub-humid climate), performing even better than seeds of *Jatropha*, that come from a similar climate as the experimental site. *Moringa* was the species with the lowest capabilities to propagate and survive due to its having the lowest germination and survival rate combined with the longest mean germination time over this and previous experiments, even though the seeds came from plants located in a similar climate (Aw). Studies over *Moringa* seed germination found that temperatures above 20 °C significantly increase mean germination time and germination rate; thus they consider that a good germination performance is highly dependent on warm temperatures. However, even with fresh seeds, the germination percentages reported by different investigations were below 90% [18,32]. During this experimental period, the site had average temperatures 4.4 °C lower than the ones recorded during the previous experiments. These temperatures could also have affected germination speed, which was slower compared to previously reported experiments for the three species. We observed that seeds of *Moringa* that did not germinate and plants that did not survive had problems with fungi. In this regard, we consider that the higher humidity and organic contents in the substratum were detrimental for *Moringa*, since it is a species adapted to very arid soils [4], and it may lack defenses against pathogens living in these environments. We also found that for the three species, heavy substrata are detrimental for a good propagation and germination speed (Table 1), especially for *Jatropha*, which registered the lowest average germination in the site. *Jatropha*’s germination problems are related to its thick testa, the thickest of the three species, and its innate vegetative rest. Nevertheless, the substratum with added compost used in this experiment aided in increasing the permeability of the testa, which helped to hydrate the embryo, resulting in an improvement of the germination rate [33]. Thus, the mix of 5:1 sand/compost in the substratum of this experiment provided nutrients to improve robustness of the sprouts, facilitating good propagation of the three species [29,33], which are mostly adapted to light soils [4,7].

Considering temperatures and substrata, in terms of the capacity of establishment, *Ricinus* was the most successful species.

### 4.2. First Phase of Seedling Growth

#### 4.2.1. Stem Growth Rates

For the first growing phase, the highest growing rates of *Ricinus* over *Jatropha* and *Moringa* were similar to the ones reported by previous experiments at the site, and the same as the highest RCD growth rates and the lowest slenderness found in *Jatropha*. This behavior confirms *Ricinus* as the most pioneering species of the three. *Jatropha*, with thicker stems and lower slender indexes, reflects a strong adaptation to dry environments, storing resources at the base of its stem [34]. It is important to remark, here, that this subtropical site has a pronounced dry season that can last up to five or six months; thus, native trees and shrubs, like *Jatropha*, have developed morphological adaptations to deal with this dry period [35]. *Moringa* represented a different strategy to cope with arid places because it developed the thinnest stem (Figure 1), but also 11% higher elongation rates than *Jatropha*. This behavior gave *Moringa* an advantage for light competition over *Jatropha*, but without expending as many resources as the other species in its aboveground development.

#### 4.2.2. Leaf Production

*Moringa* attracted attention for its highest leaf production during the last 30 days of the two months after sprouting. Previous data show that *Moringa* always produced more leaves than *Ricinus* and *Jatropha*, even in poor sandy soils (Table 2). This characteristic of *Moringa* is an inherent ability of the species linked to arid soils [4]. The compound leaves of *Moringa*, which are thinner and have lighter blades, are also related to a lower energetic investment [36]; therefore, they can be produced faster than the ones of *Ricinus* and *Jatropha*, giving *Moringa* advantages for better plant development in the long term.

#### 4.2.3. Root Systems and Their Relationships with Aboveground Growth Patterns

The longest and more numerous branches of *Ricinus* roots point towards its highest capacity to explore and obtain resources from its environment. Morphologically, *Ricinus* roots are linked to faster plant development [37]. Thus, the highly significant correlation between root volume and leaf area during its early growing phase confirmed that root development is aimed at increasing its photosynthetic capacity and, as a result, *Ricinus* had the largest leaves of the three species.

During this early phase, *Moringa* and *Jatropha* had statistically similar root volumes (P = 0.79), which were two times the volume of the roots of *Ricinus*. However, they differed in their root morphology and in the distribution of their reserves. *Moringa* stood out for its thickest napiform root since its first two months after sprouting, which represents its particular strategy to store reserves belowground which started at the first phase of growth. In this way, it can tolerate dry periods or being damaged aboveground and survive [38,39]. The highly significant correlation between leaf area and stem height indicates that *Moringa* was using its photosynthetic organs with emphasis in producing belowground reserves. On the other hand, the highest correlation between RCD and root volume in *Jatropha* indicates that this species uses its roots to search for resources and store them at the base of its stem, which is a characteristic of stem succulent species from dry environments [38].

### 4.3. Second Phase: Plant Performance after Transplanting

#### 4.3.1. Survival Rate and Aboveground Growth

The fact that the three species had 100% survival rate and very good development just 30 days after severe root pruning and bare root transplanting (Figure 2) indicate that these species are robust enough to survive and recover from transplant shock. We make this assumption because plants without transplant from previous experiments at the site, but under different soil types, reached smaller values of heights and RCDs. The higher development of the three species here can be explained by the 20% compost that integrated the substratum, which helped to increase plant size, due to its good percentage of N. Studies of *Moringa* indicate that this species can increase its biomass 1.9 times if 2.5% N is added to the soil [40], and in this experiment its height was increased by 246% and its RCD by 77%. For *Ricinus*, 4.1% N could increase plant height 1.6 times and plant biomass 2.2 times [41], and in this experiment, it increased height by 258% and RCD by 37%. For *Jatropha*, application of only 0.20% N can increase plant height 1.8 times and RCD 1.2 times [42]. These data are compatible with the highest increment in height (277%) and RCD (137%) for *Jatropha*, meaning that, aboveground, *Jatropha* obtained the highest advantage of increased soil nutrition. Thus, we recommend the amendment of compost to arid soils for good development of the three species.

Considering previous studies (Table 2), it is remarkable that even in heavy soils, *Ricinus* always obtained higher stem elongation, while *Jatropha* had the slowest growth rates. Considering that irrigation was constantly provided, *Moringa* and *Ricinus* used the available water and nutrients to accelerate their growth, as observed by the increments in their elongation and branches; instead of this strategy, *Jatropha* used these resources with a major focus in storing them at the base of its stump. This behavior can be noted because this was the species with the thickest RCD, the highest moisture content (44% more than *Moringa* and 59% more than *Ricinus*), and the densest stem of the three species. The slowest stem elongation rates and the lowest branching formation of *Jatropha* place this plant as the one with slowest development of the three species but, at the same time, it was the one with the highest dry avoidance capabilities during its early growing phase [38].

#### 4.3.2. Plant Defoliation

Plant defoliation after transplanting was caused from root loss, which decreased the ability to absorb water and nutrients and decreased the reserves of these compounds located at the roots, such as carbohydrates, minerals, and water [9]. In this regard, *Moringa* was more affected by transplant shock than *Jatropha* and *Ricinus* because *Moringa* had the highest defoliation. *Moringa* had the highest root volume, but also had higher tap root diameter than RCD, which means this species was storing more resources belowground than aboveground, contrary to *Jatropha* and *Ricinus*, which stored more reserves in their stems. Therefore, trimming of roots could more greatly affect its reserves system to avoid stressful situations, which is one strategy of this species [38], triggering leaf senescence as a result of higher nutrient deficiencies and water stress. Besides this, the thicker leaves of *Jatropha* and *Ricinus* are considered to be drought-resistant [43,44] and, therefore, they were able to resist transplant shock with lower or null defoliation, like *Jatropha*, which also had the thickest leaves. In this regard, we can remark that leaf thickness was correlated with plant defoliation, being the species with the thinnest leaves and highest defoliation percentage.

#### 4.3.3. Branching and Flowering

A comparison of the three species show that *Ricinus* was the one with the fastest development, since its plants produced the highest number of branches and reached maturity in a period similar to the one reported for this genotype at the site, which is around 150 days [28]. For the case of *Jatropha* and *Moringa*, it has been documented that they require 12 or more months to mature [45]. Therefore, the lack of flowers in both species was normal at this stage and we could not evaluate this phase during this experimental period. Nevertheless, plant growth recorded in this experiment was similar or even greater than previously reported at the site. Thus, we can assume that root trimming and transplant in these species may not cause considerable delay in reaching maturity.

### 4.4. Second Phase: Belowground Development

#### 4.4.1. Root Distribution and Root Distribution Models

Pruning roots resulted in the formation of new tap and lateral trimmed roots for the three species. *Jatropha* and *Moringa* had higher new roots than *Ricinus*, because their root diameters where thicker than the ones of *Ricinus* and, therefore, their trimmed root area for the formation of more branches was higher. The inclination angles (greater than −60°) of the new tap roots in the three species indicate that these new branches were acting as anchors to increase plant stability [16]. The lower inclination angles of *Moringa* roots can be associated with their thicker size, although these inclination angles are high enough to perform an anchorage role [20]. Consequently, root pruning did not decrease anchoring capabilities, with the increment in the number of roots being beneficial for their structure, as we theorized before the experiment.

In the horizontal distribution of roots, *Moringa* stood out for its highest volume distribution (74%) along the first 5 cm around the stump (Figure 5), which is the result of the napiform shape of its tap and lateral roots. The vertical root distribution of *Moringa* had a very characteristic distribution along the soil, resembling a Gaussian curve, which was best represented by a logistic model explaining 85% of variation, where the highest volume (31%) was located 10 cm below the surface because this species locates its reserves at a certain distance belowground as a safety measure to avoid damage [14].

*Jatropha* distributed 50% of its root volume in the first 5 cm along its stump horizontally, and 68% vertically. Root volume decay along the vertical surface was slightly lower than *Moringa*, but higher than *Ricinus*, and the logistic model was able to explain 99% of variation.

*Ricinus* distributed only 34% of its volume in the first 5 cm below the surface, and 27% in the first 5 cm around its stump. This species had the smoothest vertical and horizontal distribution, and the models representing this performance had the slowest decay speed, especially along the horizontal distribution.

Similar horizontal and vertical root distribution models were obtained for one-year-old plants of *Moringa* and *Ricinus* growing under a clay soil [14]. This means that these *Moringa* and *Ricinus* seedlings show analogous distribution patterns as adult plants without transplant. Therefore, we consider that trimming of roots did not substantially change the natural behavior of their root distribution. We did not find any model for *Jatropha* and, therefore, this is the first model to be presented; nevertheless, this study has found that *Jatropha* has a root distribution similar to *Ricinus*, while *Moringa* follows a completely different pattern.

#### 4.4.2. Root Trays and Their Implications for Aboveground Development

Root trays indicate that the highest capacity to cover soil along horizontal and vertical surfaces was obtained by *Ricinus*, which has the highest capacity of soil exploration and cohesion [31], followed by *Jatropha* and, with the lowest capacity, *Moringa*. Therefore, we consider that *Ricinus* could be more successful than *Jatropha* or *Moringa* if transplanted in eroded or loose soils. The higher number and longer extension of *Ricinus* lateral roots indicate its greater capacity to recover more quickly from transplant compared to the others. Small diameters and considerably longer and numerous roots are traits associated with maintaining plant productivity under drought stress, because longer and thinner roots are able to reach more resources for a lower energy investment [36]. This performance can be observed in the plant weight distribution, because *Ricinus* had the lightest fresh and dry weight roots of the three species, maintaining similar fresh and dry weights of stem compared to *Moringa*, which had the heaviest roots. Prior to transplant, the three species had the same root trays, with *Ricinus* having the most extensive root system, *Jatropha* moderate, and *Moringa* the most compact. Therefore, we considered that the natural root distribution patterns of the three species did not change with root pruning and transplant.

In the second phase, the root volume of *Ricinus* correlations more strongly with stem height and RCD than leaf area, which did not significantly correlate with root parameters. This indicates that stem elongation is highly dependent on root development. We found that plants with thicker tap roots had significant correlations with the number of branches, and the plants with longer roots were the ones that were flowering earlier; therefore, we can state that these root trays play an important role in plant maturation for *Ricinus* [28].

*Jatropha* followed *Ricinus* in root covering, but it developed only 29% of the number of root branches of *Ricinus*, although it had 58% more branches than *Moringa*, and almost three times longer roots than *Moringa*. Like *Ricinus*, *Jatropha* produced more biomass in its stem than in its roots, but these roots were heavier and thicker compared to *Ricinus*. Thus, we state that *Jatropha* benefited from root pruning because this species usually does not develop highly branched roots during its juvenile stage [11], and the number of roots increased more than six times after this treatment. Other research of three-year-old plants indicate that this species can grow fine roots of greater lengths in the surface soil [46], same as are found in juvenile plants, but it has a conservative root system, which means that these roots will not be as good as the ones of *Ricinus* at capturing nutrients and water; consequently, plant development of *Jatropha* is slower. However, *Jatropha* had the strongest relationship between root volume and RCD and number of leaves, which means that this species will depend more on its root development than the other two species to store resources at the base of its stem and to increase its photosynthetic capabilities.

*Moringa*, with the thickest and shortest roots, was the species with the lowest soil covering capacity, although it developed the longest thicker roots and the heaviest root system, with even longer tap elongation at the trimmed tip than *Jatropha*. Correlations between below- and aboveground parameters indicate that, during the second growth phase, *Moringa* initiated storing resources at the base of its stump as a stem caudiciform species [38], similarly to *Jatropha* since its early phase. Moreover, *Moringa* allocated more water and biomass to its roots than its other organs, which is a typical characteristic of the species [14]. Therefore, we consider that transplanting did not affect its natural biomass distribution. In addition, by the end of the experimental period, the three species were developing faster than other ones without transplant reported at the site. Thus, we can say that after the recovery time, increases in root density had a positive influence over plant development.

#### 4.4.3. Above- and Belowground Biomass Estimation by Allometric Data

The models obtained in this experiment indicate that allometric relationships differed among species, with each one depending on different allometric measurements for estimated biomass fractions (leaves, stem, and roots). Studies with small trees and shrubs have found that height and RCD are the main parameters used to estimate aboveground biomass [24], as we have confirmed in this study in the case of *Ricinus*, whose equations for estimating stem and root biomass involved height and RCD; nevertheless, in the case of *Moringa* and *Jatropha*, we must to consider that plants at this juvenile stage do not have the same biomass partitions as their adults and, therefore, leaf area played a more important role for their biomass estimation.

In adult trees of *Moringa*, the RCD instead of the height of the stem is a better estimator for biomass partitions; however, for adult *Ricinus*, height and RCD are used together to estimate biomass [14], while for *Jatropha*, crown diameter, height, and RCD are required [17]. Thus, we consider that for *Ricinus*, the size of the stem is the most important characteristic for estimating biomass during its early development, while in *Jatropha*, leaf data is the most important parameter to estimate biomass. Considering weight distribution, we noticed that *Jatropha* had the lowest investment in root biomass for develop the highest stem biomass. This relationship can be a consequence of *Jatropha* being a stem succulent species [44], which depends more on its stem to avoid drought than *Moringa* or *Ricinus*. The allometric relationships of the three species indicate that they maintained their natural biomass partition after pruning and transplant, recovering from the shock in less than 83 days, which is a shorter time compared with other perennial species that required more than 100 days [11].

## 5. Conclusions

Among the three oily species, *Ricinus* was the species with the highest capacity to survive and propagate in a subtropical climate with light substratum.

Trimming of roots increased plant stability in the soil because it induced higher root branching in the three species, especially for *Jatropha* and *Moringa*, which normally develop a low number of branches during the early stage.

Root pruning did not affect plant behavior, with *Ricinus* being the species with the most efficient root system, characterized by long thin roots linked to the fastest shoot development before and after transplant. *Moringa*, with the thickest and shortest roots, maintained its strategy to store reserves belowground and at the base of its stump, while *Jatropha*, with shorter and thicker roots than *Ricinus*, was the species with the slowest aboveground development, focusing mostly in storing reserves at the base of its stump.

The allometric models indicated that the best predictor for *Jatropha*’s biomass was its root collar diameter, while the best estimators for *Ricinus* and *Moringa* were their stem heights and leaf area.

Among the three, *Ricinus* was the earliest successional species, followed by *Moringa*, and *Jatropha* was the most recent successional species.

## Figures and Tables

**Figure 1 plants-08-00258-f001:**
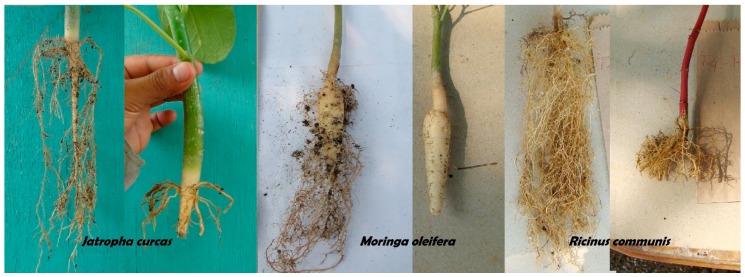
Roots of 63-day-old *Jatropha curcas*, *Moringa oleifera*, and *Ricinus communis* before and after pruning.

**Figure 2 plants-08-00258-f002:**
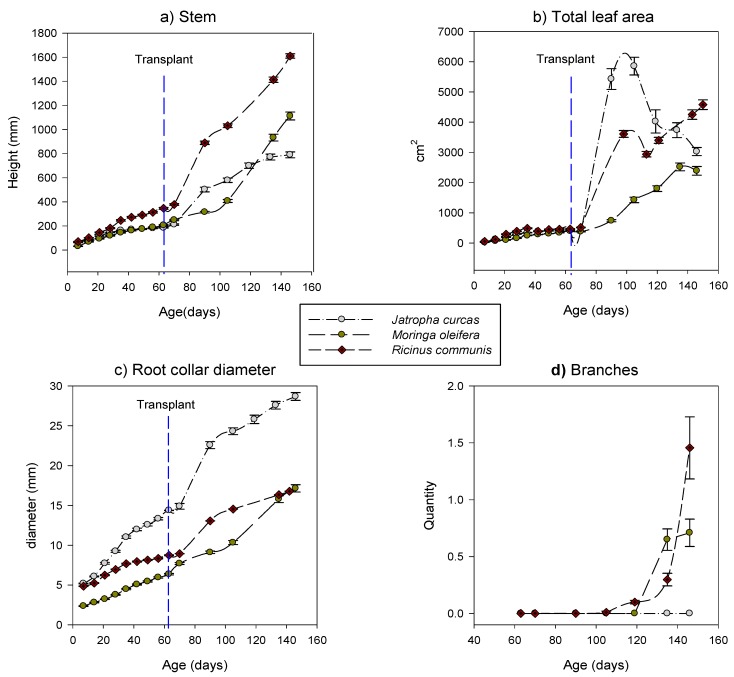
Average growth curves of *Jatropha curcas*, *Moringa oleifera*, and *Ricinus communis* seedlings. (**a**) Stem height; (**b**) the total leaf area represents the area of the youngest mature leaf by the number of leaves; (**c**) diameter at the base of the stem; and (**d**) branch production. None of the species registered branches before transplant; thus, the dashed vertical line representing the day of transplant was not drawn for this variable at (**d**). Error bars represent the standard error of the mean.

**Figure 3 plants-08-00258-f003:**
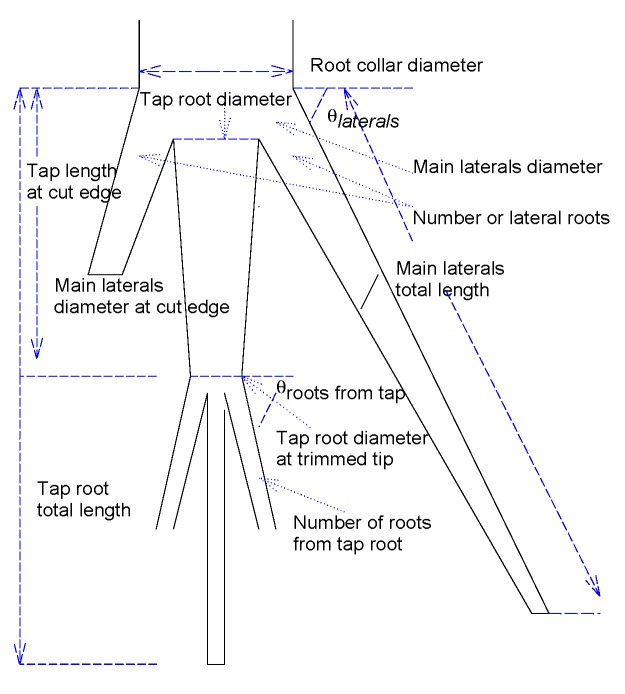
Root measurements taken at tap and main lateral roots indicated in Table 4.

**Figure 4 plants-08-00258-f004:**
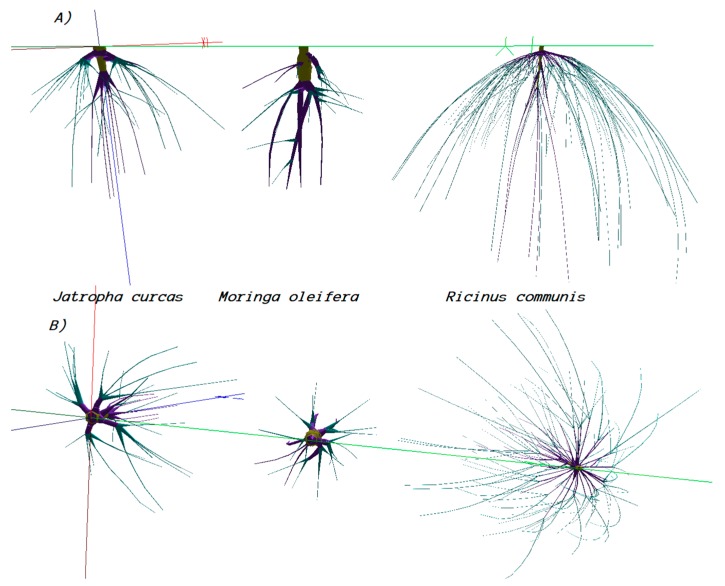
Roots of *Jatropha*, *Moringa*, and *Ricinus* digitized 83 days after their root pruning and transplant. (**A**) Vertical projection; (**B**) Horizontal projection.

**Figure 5 plants-08-00258-f005:**
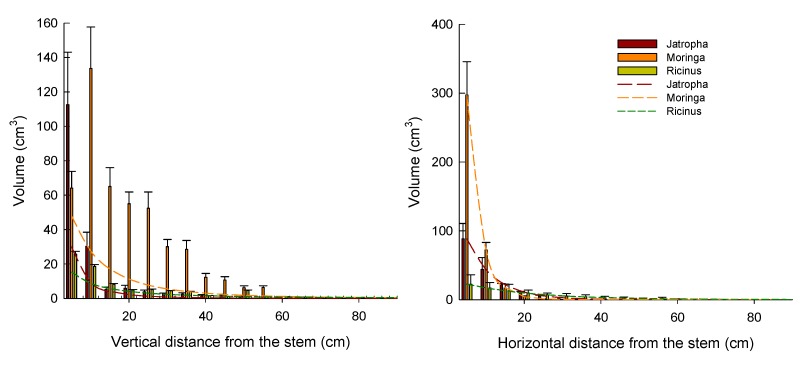
Root volume distribution of *Jatropha*, *Moringa*, and *Ricinus* 83 days after root pruning and transplanting. Vertical error bars represent the standard error of the mean. The dashed lines represent the estimated models.

**Table 1 plants-08-00258-t001:** Percentage and average germination days of *Moringa oleifera*, *Jatropha curcas*, and *Ricinus communis* obtained in this and previous experiments at the experimental site.

Species	*Moringa*			*Ricinus*			*Jatropha*		
Variable	Exp1	Exp2	This experiment	Exp1	Exp2	This experiment	Exp3	Exp3	This experiment
Germination percentage	95.0	60.0	86.0	96.0	80.0	100.0	77.0	25.0	97.0
Mean germination time (days)	5.9	13.0	14.1	4.7	10.0	7.5	6.2	10.8	9.1
Substratum	Sand	Clay	S-C (5:1) ^†^	Sand	Clay	S-C (5:1) ^†^	Sand	Clay	S-C (5:1) ^†^
Average Temperature °C)	27.0	25.5	23.2	28.7	25.5	22.2	27.4	27.4	23.2
Humidity (%)	88.7	78.7	73.9	88.5	78.7	71.3	73.8	73.8	73.3

Exp1 [27]; Exp2: [28]; Exp3: [29]. This experiment: ^†^ Sand/Compost 5:1.

**Table 2 plants-08-00258-t002:** Mean growth rates (mm/day) and leaf production (leaf/day) ± SE of *Moringa oleifera*, *Jatropha curcas*, and *Ricinus communis*, and environmental conditions registered in this and previous experiments conducted at the experimental site.

Variable	*Moringa*		*Ricinus*		*Jatropha*	
	Exp2	This experiment	Exp2	This experiment	Exp1	This experiment
Plant age (days)	45	63	45	63	45	63
Height	4.66 ± 0.09	3.30 ± 0.07	5.08 ± 0.09	5.49 ± 0.12	4.22 ± 0.12	2.93 ± 0.05
Root collar diameter	0.09 ± 0.00	0.10 ± 0.00	0.14 ± 0.00	0.14 ± 0.00	0.19 ± 0.02	0.23 ± 0.00
Leaf production	0.11 ± 0.00	0.16 ± 0.00	0.07 ± 0.00	0.07 ± 0.00	0.06 ± 0.00	0.11 ± 0.01
Substratum	Sand	S-C (5:1) ^†^	Sand	S-C (5:1)	Sand	S-C (5:1)
Temperature (°C)	26.01 ± 1.17	24.69 ± 2.27	26.01 ± 1.17	24.69 ± 2.27	24.69 ± 1.45	24.69 ± 2.27
Humidity (%)	88.54 ±3.03	75.00 ± 5.90	88.54 ± 3.03	75.00 ± 5.90	75.00 ± 6.61	75.00 ± 5.90

Exp1 [16], Exp2 [14], ^†^ Sand/Compost 5:1.

**Table 3 plants-08-00258-t003:** Average root dimensions ± SE of 63-day-old *Moringa oleifera*, *Jatropha curcas*, and *Ricinus communis*.

Parameter (n = 100)	*Moringa*	*Ricinus*	*Jatropha*
Tap root length (mm)	173.59 ± 4.20 **c**	411.14 ± 5.92 **a**	280.40 ± 6.14 **b**
Main lateral roots length (mm)	196.25 ± 4.79 **c**	351.60 ± 4.17 **a**	289.00 ± 5.24 **b**
Maximum tap root diameter (mm)	18.5 ± 0.35 **a**	6.09 ± 0.12 **c**	9.71 ± 0.29 **b**
Main lateral roots diameter (mm)	1.85 ± 0.13 **b**	1.05 ± 0.02 **c**	3.97 ± 0.17 **a**
Number of lateral roots	1.09 ± 0.03 **c**	25.18 ± 0.48 **a**	5.15 ± 0.28 **b**
Tap root + lateral roots volume (cm^3^)	23.82 ± 1.69 **a**	11.11 ± 0.29 **b**	23.33 ± 1.52 **a**
Tap root diameter/Main lateral roots diameter	12.53 ± 0.54 **a**	5.90 ± 0.13 **b**	2.5 ± 0.10 **c**
Tap root volume/laterals root volume	19.88 ± 1.14 **a**	1.29 ± 0.07 **b**	1.40 ± 0.14 **b**

Different letters in the same row indicate statistical differences (P < 0.05).

**Table 4 plants-08-00258-t004:** Linear correlations between above- and belowground data of *Moringa oleifera*, *Jatropha curcas* and *Ricinus communis* seedlings 63 days after germination.

Species (n = 100)	Parameter	Height	RCD	Number of Leaves	Leaf Area of the Youngest Fully Mature Leaf
***Jatropha***	Tap root length	0.48 **	0.51 **	0.34 *	0.22
	Laterals length	0.49 **	0.52 **	0.24	0.23
	Tap root diameter	0.44 **	0.54 **	0.10	0.26
	Lateral roots diameter	0.40 **	0.57 **	0.18	0.12
	Number of Laterals	0.16	0.15	−0.11	−0.03
	Root volume	0.62 **	0.64 **	0.17	0.21
***Moringa***	Tap root length	0.18	0.01	−0.06	0.16
	Lateral roots length	0.31 **	0.09	−0.11	0.31 **
	Tap root diameter	0.29 **	0.44 **	0.02	0.14
	Lateral roots diameter	0.33 **	0.40 **	0.03	0.35 **
	Number of lateral roots	0.10	0.07	−0.06	0.13
	Root volume	0.66 **	0.43 **	−0.06	0.59 **
***Ricinus***	Tap root length	0.30 *	0.40 **	0.06	0.08
	Lateral roots length	0.23 *	0.39 **	0.16	0.11
	Tap root diameter	0.29 **	0.42 **	−0.07	0.05
	Lateral roots diameter	−0.09	−0.11	−0.17	0.20*
	Number of lateral roots	0.26 **	0.08	0.07	−0.16
	Root volume	0.37 **	0.40 **	−0.07	0.70 **

* Significant at 0.05; * significant at 0.01; RCD = root collar diameter.

**Table 5 plants-08-00258-t005:** Mean (± SD) aboveground data of *Moringa oleifera*, *Ricinus communis*, and *Jatropha curcas*, and their environmental conditions documented during previous research at the site.

Parameter	*Moringa* ^1^	*Ricinus* ^1^	*Jatropha* ^2^
Plant height (mm)	321.4 ± 158.0	450.0 ± 249.7	209.4 ± 26.6
RCD	9.7 ± 4.1	12.27 ± 7.0	12.1 ± 1.6
Number of leaves	NA	NA	0.5 ± 0.7
Area of average leaf	NA	NA	4.84 ± 0.21
Number of branches	NA	NA	0.0 ± 0.0
Plant age (days)	103	103	110
Soil type	Clay	Clay	Sandy
Growth conditions	Open field	Open field	Plastic bag (25 × 60 cm)
Temperature °C	24.2 ± 1.5	24.2 ± 1.5	24.3 ± 2.7
Humidity (%)	78.4 ± 5.9	78.4 ± 5.9	74.3 ± 12.6

^1^ [28], ^2^ [30], NA = data not available.

**Table 6 plants-08-00258-t006:** Average root dimensions (mm) ± SE of *Moringa oleifera*, *Ricinus communis*, and *Jatropha curcas* 83 days after root pruning and transplant.

Dimension (n = 20)	*Moringa*	*Ricinus*	*Jatropha*
Tap root length at trimmed tip	89.69 ± 3.42 **b**	123.84 ± 10.63 **a**	64.38 ± 2.67 **c**
Tap root total length	409.25 ± 12.82 **b**	831.9 ± 32.36 **a**	478.50 ± 14.10 **b**
Tap root diameter	58.74 ± 1.03 **a**	10.01 ± 0.42 **c**	28.15 ± 0.76 **b**
Tap root diameter at trimmed tip	41.25 ± 1.96 **a**	5.35 ± 0.34 **c**	21.61 ± 1.10 **b**
Number of roots from tap root	3.44 ± 0.21 **c**	4.60 ± 0.40 **b**	6.10 ± 0.32 **a**
Tap root inclination (θ roots from tap)	−60.20 ± 12.19 **b**	−81.00 ± 1.81 **a**	−83.71 ± 1.09 **a**
Number of main lateral roots	2.00 ± 0.24 **c**	33.37 ± 2.56 **a**	5.60 ± 0.15 **b**
Main lateral roots total length	395.19 ± 12.53 **b**	758.58 ± 33.49 **a**	484.55 ± 22.13 **b**
Main lateral roots diameter	22.40 ± 1.16 **a**	2.45 ± 0.12 **c**	16.74 ± 0.78 **b**
Main lateral roots diameter at trimmed tip	16.78 ± 1.05 **a**	1.03 ± 0.04 **c**	10.64 ± 0.53 **b**
Lateral root inclination (θ laterals)	−53.33 ± 4.90 **a**	−24.25 ± 2.99 **b**	−31.50 ± 5.34 **b**
Tap root length/stem length	0.32 ± 0.01 **b**	0.66 ± 0.02 **a**	0.60 ± 0.01 **a**
Total tap root length/total lateral lengths	1.04 ± 0.02 **a**	1.11 ± 0.04 **a**	1.00 ± 0.03 **a**
Root collar diameter/tap root diameter	0.66 ± 0.02 **b**	1.63 ± 0.06 **a**	1.41 ± 0.03 **a**
Tap root diameter/main lateral roots diameter	2.84 ± 0.13 **a**	4.28 ± 0.27 **a**	1.72 ± 0.05 **b**
Tap root + main lateral roots volume (cm^3^)	275.69 ± 19.22 **a**	76.89 ± 5.97 **c**	218.24 ± 27.50 **b**

Different letters in the same row indicate statistical differences (P < 0.05).

**Table 7 plants-08-00258-t007:** Linear correlations between above- and belowground data of *Jatropha curcas*, *Ricinus communis*, and *Moringa oleifera* 83 days after root pruning and transplant.

Species (n = 20)	Parameter	Height	RCD	Number of Leaves	Leaf Area	Number of Branches
***Jatropha***	Tap root length	0.61 **	0.58 **	0.48 *	0.32 *	--
	Lateral roots length	0.42 *	0.65 **	0.44 *	0.50 **	--
	Tap root diameter	0.70 **	0.81 **	0.73 **	0.12	--
	Lateral roots diameter	0.50 *	0.63 **	0.61 **	0.05	--
	Number of Laterals	0.46 *	0.44 *	0.35 *	0.37 *	--
	Root volume	0.67 **	0.84	0.69 **	0.25	--
***Moringa***	Tap root length	0.55 **	0.66 **	0.44 *	0.29	0.18
	Lateral roots length	0.40 *	0.53 **	0.20	0.26	0.05
	Tap root diameter	0.76 **	0.80 **	0.55 **	0.42 *	0.01
	Lateral roots diameter	0.13	0.15	0.27	−0.21	0.21
	Number of Laterals	0.66 **	0.76 **	0.58 **	0.21	0.20
	Root volume	0.76 **	0.76 **	0.59 **	0.32	0.12
***Ricinus***	Tap root length	0.57 **	0.23	0.25	−0.20	0.09
	Lateral roots length	0.30 *	0.22	0.30 *	−0.13	0.12
	Tap root diameter	0.03	0.51 **	0.16	0.00	0.26*
	Lateral roots diameter	0.34 **	0.21	0.12	0.06	−0.07
	Number of lateral roots	0.06	−0.12	−0.08	0.17	−0.02
	Root volume	0.43 **	0.42 **	0.24	0.15	0.10

** Significant at 0.05; * significant at 0.01. -- The stem of *Jatropha* did not develop branches.

**Table 8 plants-08-00258-t008:** Root distribution models of *Jatropha curcas*, *Moringa oleifera*, and *Ricinus communis* 83 days after root pruning and transplanting.

Parameters and Model		*Jatropha*	*Moringa*	*Ricinus*
Vertical distribution	Vertical root volume (cm^3^) at a distance D	$\frac{181.20}{1+{\left(\frac{D}{58.77}\right)}^{3.09}}$	$\frac{92.64}{1+{\left(\frac{D}{248.60}\right)}^{3.59}}$	$\frac{33.58}{1+{\left(\frac{D}{97.22}\right)}^{1.91}}$
	SS	37.64	3195.32	26.40
	r^2^	0.99	0.85	0.97
Horizontal distribution	Horizontal root volume (cm^3^) at a distance D	$178.50e^{-0.014D}$	$676.60e^{-0.02D}$	$29.88e^{-0.006D}$
	SS	710.21	1.38	26.35
	r^2^	0.93	0.99	0.96

D = distance from the base of the stump (cm); r^2^ = coefficient of determination; SS = square sum.

**Table 9 plants-08-00258-t009:** Average (± SE) fresh and dry weight (g) of *Moringa oleifera*, *Ricinus communis*, and *Jatropha curcas* 83 days after root pruning and transplanting.

Parameter (n = 20)	*Moringa*	*Ricinus*	*Jatropha*
Leaf fresh weight	63.72 ± 7.70 **b**	74.83 ± 3.41 **a**	75.10 ± 1.85 **a**
Stem fresh weight	127.19 ± 12.76 **b**	136.00 ± 5.06 **b**	187.91 ± 5.78 **a**
Root fresh weight	136.78 ± 11.85 **a**	75.16 ± 3.87 **b**	59.03 ± 1.83 **c**
Above/below ground fresh weight	1.42 ± 0.07 **c**	2.92 ± 0.37 **b**	4.84 ± 0.21 **a**
Leaf dry weight	12.07 ± 1.40 **b**	9.37 ± 0.41 **c**	14.10 ± 0.12 **a**
Stem dry weight	26.48 ± 2.12 **b**	20.17 ± 0.77 **b**	81.67 ± 3.29 **a**
Root dry weight	19.69 ± 1.89 **a**	7.54 ± 0.37 **c**	12.54 ± 0.40 **b**
Above/below ground dry weight	2.14 ± 0.15 **c**	4.01 ± 0.15 **b**	7.65 ± 0.18 **a**

Different letters in the same row indicate statistical differences (P < 0.05).

**Table 10 plants-08-00258-t010:** Biomass estimation of *Moringa*, *Ricinus*, and *Jatropha* by their allometric relationships.

Biomass (mg)	*Moringa*	*Ricinus*	*Jatropha*
**L = Leaf**			
Variables	HNA = H N A	HNA = H N A	NA = N A
Model	L = −1.8 + 10.3*ln* (HNA)	L = 15.3(HNA)^0.6^	L = 17.7 + 3.8*ln* (NA)
Type	Logarithm	Power	Logarithm
r^2^	0.83	0.67	0.57
SS	106.6	47.8	9.8
**S = Stem**			
Variables	HNA = H N A	DH = D H	D
Model	S = 5.4 + 15.8*ln* (HNA)	S = 540(DH)^0.84^	S = 267.1 + 59.6ln(D)
Type	Logarithm	Power	Logarithm
r^2^	0.87	0.70	0.54
SS	187.5	138.9	1733.2
**R = Root**			
Variables	HNA = H N A	DH = D H	DNA = D N A
Model	R = 2.9 + 12.7*ln*(HNA)	R = 182.6(DH)^0.81^	R = 67.1DNA^0.40^
Type	Logarithm	Exponential	Power
r^2^	0.71	0.41	0.57
SS	328.6	61.9	36.1

H = height (m); D = RCD (m); N = Number of leaves; A = Area of average leaf (m^2^); r^2^ = coefficient of determination; SS = square sum.

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
