# Peer review of "Seedling Characteristics of Three Oily Species before and after Root Pruning and Transplant"

_plants, 2019, doi:10.3390/plants8080258_

Round 1
Reviewer 1 Report
This revised manuscript is an improvement over the original version. The Introduction explains the aim of the study much better. Now, the paper is clear and easy to read. Materials and methods are adequate and sufficiently described. Data and results are presented appropriately. The discussion is well written and the conclusions clearly summarize the obtained results.
There are some typos in the manuscript
Line 25, change Jatropha to Jatropha
Line 83, change Analyze to analyze
Line 527, change moringa to Moringa
Author Response
Response to Reviewer 1 Comments
Line 25, change Jatropha to Jatropha
All the words were checked for italic formats
Line 83, change Analyze to analyze
All the words were checked for italic formats
Line 527, change moringa to Moringa
All the words were checked for italic formats
Reviewer 2 Report
The manuscript aiming to evaluate propagation and growth characters of three oily species in subtropical climate, and to compare the above and below ground relationship before and after root pruning, and to analize their root structures. I don't understand why do you evaluate these characters in the same experimental design? Do you compare the three species? Why?
I think that you have to better explain the aim of this comparative study.
Author Response
1) The manuscript aiming to evaluate propagation and growth characters of three oily species in subtropical climate, and to compare the above and below ground relationship before and after root pruning, and to analize their root structures.
I don't understand why do you evaluate these characters in the same experimental design?
We want to evaluate above and below structures before and after pruning because there are not previous data comparing plant responses in these three species. Therefore we needed to perform the evaluation in the same experimental design.
2) Do you compare the three species? Why?
Since each species has a different morphology and strategy to deal with arid conditions, pruning roots is a way to obtain their maximum response to hydric stress. And we wanted to learn which one has the best response under this subtropical environment because this is the environment where the three of them are recommended to be cultivated.
3) I think that you have to better explain the aim of this comparative study.
We have added a better explanation in lines 42-44; and lines 76 to 80.
We also had added our main research questions in lines 80-82.